# Cooking/Window Opening and Associated Increases of Indoor PM$_{2.5}$ and NO$_2$ Concentrations of Children's Houses in Kaohsiung, Taiwan

**Yu-Chuan Yen [1], Chun-Yuh Yang [1], Kristina Dawn Mena [2], Yu-Ting Cheng [1] and Pei-Shih Chen [1,3,4,5,*]**

[1] Department of Public Health, College of Health Science, Kaohsiung Medical University, Kaohsiung City 807, Taiwan; ponder10003@gmail.com (Y.-C.Y.); chunyuh@kmu.edu.tw (C.-Y.Y.); llssdog@hotmail.com (Y.-T.C.)

[2] Epidemiology, Human Genetics, and Environmental Sciences, School of Public Health, University of Texas Health Science Center at Houston, Houston, TX 77046, USA; Kristina.D.Mena@uth.tmc.edu

[3] Institute of Environmental Engineering, College of Engineering, National Sun Yat-Sen University, Kaohsiung City 807, Taiwan

[4] Department of Medical Research, Kaohsiung Medical University Hospital, Kaohsiung City 807, Taiwan

[5] Research Center for Environmental Medicine, Kaohsiung Medical University, Kaohsiung City 807, Taiwan

\* Correspondence: pschen@kmu.edu.tw; Tel.: +886-7-3110811

**Abstract:** High concentrations of air pollutants and increased morbidity and mortality rates are found in industrial areas, especially for the susceptible group, children; however, most studies use atmospheric dispersion modeling to estimate household air pollutants. Therefore, the aim of this study was to assess the indoor air quality, e.g., CO, CO$_2$, NO$_2$, SO$_2$, O$_3$, particulate matter with aerodynamic diameter less than 2.5 μm (PM$_{2.5}$), and their influence factors in children's homes in an industrial city. Children in the "general school", "traffic school", and "industrial school" were randomly and proportionally selected. Air pollutants were sampled for 24 h in the living rooms and on the balcony of their houses and questionnaires of time–microenvironment–activity-diary were recorded. The indoor CO concentration of the traffic area was significantly higher than that of the industrial area and the general area. In regard to the effects of window opening, household NO$_2$ and PM$_{2.5}$ concentrations during window opening periods were significantly higher than of the reference periods. For the influence of cooking, indoor CO$_2$, NO$_2$, and PM$_{2.5}$ levels during the cooking periods were significantly higher than that of the reference periods. The indoor air quality of children in industrial cities were affected by residential areas and household activities.

**Keywords:** indoor air quality; children's house; industrial city; window opening; cooking

## 1. Introduction

According to the Environmental White Paper of Taiwan Environmental Protection Agency (Taiwan EPA), the annual average concentrations of ambient CO, NO$_2$, SO$_2$, and O$_3$ in 2008 were 0.47 ppm, 16.90 ppb, 4.35 ppb, and 29.09 ppb, respectively. The Kaohsiung–Pingtung area was the worse polluted area in Taiwan and accounted for 5.93% of station-days of the Pollutant Standards Index (PSI) > 100. Especially, Kaohsiung is a heavy industrial city. In industrial areas, high concentrations of air pollutants and increased morbidity and mortality rates are found, depending on the types of industrial activities and exposure concentrations in residential areas [1,2]. Children are more susceptible to the health effects of air pollution than adults due to not having full development of their pulmonary metabolic capacity [3]. Long-term exposure of air pollution may affect children's lung development [4]. Previously, most of the studies revealed that ambient pollution such as particulate

matter with aerodynamic diameter less than 10 and 2.5 μm ($PM_{10}$ and $PM_{2.5}$), sulfur dioxide ($SO_2$), nitrogen dioxide ($NO_2$), volatile organic compounds (VOCs), etc. in industrial areas may increase the risk of respiratory symptoms, and attacks of asthma in children [1,5,6]. Therefore, indoor air quality of children's homes may be very important to children's health, especially in industrial cities, since children spend most of their time at home [7].

Indoor air quality may be affected by indoor human activities such as cooking, smoking, cleaning, etc. and the infiltration of outdoor pollutants produced from the traffic or industrial sources [8–11]. For example, $SO_2$, $NO_x$, $PM_{2.5}$, and carbon monoxide (CO), the major conventional air pollutant in steel plants, oil refineries, and vehicular exhaust emissions [12–14], may enter a house through cracks and windows [15,16]. In addition, if the indoor air is not well ventilated, the air pollutants may accumulate in the indoor environment, and then seriously affects the health of the inhabitants [17].

Previously, atmospheric dispersion modeling was used to estimate the household concentrations of indoor air pollutants in industrial areas [1,18,19]. Only a few studies actually measured individual exposure [20] and household concentrations [21–23], and these studies only focused on PM mass concentrations, elemental composition, and VOCs concentrations. However, other air pollutants e.g., CO, carbon dioxide ($CO_2$), $NO_2$, $SO_2$, and ozone ($O_3$) in households in industrial cities also need to be considered. Therefore, the main aim of this study was to assess the indoor air quality including CO, $CO_2$, $NO_2$, $SO_2$, $O_3$, and $PM_{2.5}$, temperature and relative humidity, and their influence factors (e.g., window opening and cooking) in children's homes in an industrial city—Kaohsiung City. To our knowledge, this is the first study to assess the indoor air quality including CO, $CO_2$, $NO_2$, $SO_2$, and $O_3$ in children's homes in an industrial city. In addition, the second aim was to evaluate potential determinants of indoor air pollutants levels of occupants' activities, including cooking and window opening, etc. It is also the first study to reveal the differences of air pollutants between cooking periods/window opening periods and reference periods through a time–microenvironment–activity-diary via a questionnaire in one-hour time segments.

## 2. Materials and Methods

### 2.1. Study Area

Kaohsiung City (22°38′ N, 120°17′ E), located in southern Taiwan and with the population density of 9962.6/$km^2$ in 2010, is the largest industrialized harbor city in Taiwan with intense traffic and heavy industries including the largest steel plant (the China Steel Corporation, which also ranked the 19th steel mill in the world in 2005), the largest oil refinery (the CPC Corporation), the largest international shipbuilding (it ranked 6th in the world in 2005) in Taiwan, and many petrochemical industries.

### 2.2. Study Design

In April 2010, we selected three elementary schools in Kaohsiung City. One elementary school had a general air quality monitoring station of Taiwan EPA on the roof of the 4th floor, so we called this school a "general school". Another elementary school was 0.33 km from Taiwan EPA's traffic air quality monitoring station and was regarded as a "traffic school". The "industrial school" was an elementary school located near the Xiaogang Industrial Zone in Kaohsiung City and about 0.30 km from Taiwan EPA's air monitoring station. The study population was limited to children who attended these schools. The number of students in the "general school", "traffic school", and "industrial school" were 1669, 987, and 960, respectively. After obtaining the assented of the child and the permission of the parents, we recorded the subjects who agreed to home visits for environmental sampling. Children were randomly and proportionally selected from each school to participate in this study. Finally, the home visits of 32, 16, and 12 participants in the "general school", "traffic school", and "industrial school", respectively, were completed between April 2010 and October 2010.

*2.3. Air Sampling*

Indoor air pollutants including CO, $CO_2$, $NO_2$, $SO_2$, $O_3$, $PM_{2.5}$, temperature, and relative humidity were measured by real-time monitoring equipment for 24 h in the living rooms. We also measured the atmospheric CO, $CO_2$, $NO_2$, $SO_2$, $O_3$, and $PM_{2.5}$ on the balcony as outdoor concentrations. All instruments were placed on the bench at a height of approximately 1 m above the ground. The PM was measured by a real-time optical scattering instrument (DUSTTRAK™ DRX Aerosol Monitor Models 8533, TSI Incorporated, Shoreview, MN, USA) and the measurements were taken every 1 s by the flow rate of 3.0 L/min with detectable concentration from 0.001 to 150 $mg/m^3$. The CO, $CO_2$, $NO_2$, $SO_2$, indoor temperature, and relative humidity were also recorded (KD-airboxx, KD Engineering, Blaine, WA, USA) every 15 s with the measuring range of 0 to 500 ppm, 0 to 10,000 ppm, 0 to 20 ppm, 0 to 20 ppm, 0 to 50 °C, and 5% to 95%, respectively. The accuracy of CO, $CO_2$, $NO_2$, and $SO_2$ were ±3% of reading or 2 ppm (whichever was greater), ±5% of reading or 60 ppm (whichever was greater), 0.25 ppm, and 0.25 ppm, respectively. The resolution of CO, $CO_2$, $NO_2$, and $SO_2$ were 0.1 ppm, 1 ppm, 0.01 ppm, and 0.01 ppm, respectively. In terms of $O_3$, it was detected by a real-time monitoring (Model 202 Ozone monitor™, 2B Technologies Inc, Boulder, CO, USA) every 5 min with the measuring range of 0 to 250 ppm.

All real-time monitors were manufacturer-calibrated for the study in the beginning of this study and every six months. Before every field sampling, the DUSTTRAK™ DRX Aerosol Monitor Models 8533 was calibrated using emery oil aerosol and nominally adjusted to the respirable mass of standard ISO 12103-1, A1 test dust, (Arizona Dust); and the KD-airboxx, the Model 202 Ozone monitor™ were calibrated using zero gas and span gas. In addition, the zero calibrators of instruments were carried out, and the flow rate of sampling pump also was adjusted by Gilian Gilibrator-2NIOSH Primary Standard Air Flow Calibrator (Sensidyne, St. Petersburg, FL, USA) before every household sampling.

*2.4. Household Characteristics*

In addition, household characteristics including the number of occupants, air-conditioning use, smoking, incense burning, etc. were also recorded in the questionnaires. In addition to household characteristics, data on potential determinants of indoor air pollutants levels of occupants' activities, including cooking and window opening, etc. were obtained through a time–microenvironment–activity-diary via a questionnaire in one-hour time segments. We also actually evaluated the effects of window opening and cooking on indoor air pollutants. The window opening periods were defined from a time–microenvironment–activity-diary and two one-hour periods before and after window opening periods were defined as the reference periods. In terms of cooking, the cooking periods were the periods recorded by participants as cooking from a time–microenvironment–activity-diary and the reference periods were defined as the one-hour periods before the cooking periods.

*2.5. Ethics*

This study was approved by the Institutional Review Board of the Kaohsiung Medical University Chung-Ho Memorial Hospital (the protocol number was KMU-IRB-990045). Informed written consent was obtained from each child (the phonetic version of the consent form that the children read and signed) and their legal guardians.

*2.6. Statistical Analyses*

Statistical analyses in this study were performed using SAS version 9.3 (SAS Institute of Taiwan Ltd, Taipei, Taiwan). Descriptive statistics were used to describe the 24-hour of average of exposure data (indoor/outdoor air pollutant concentrations, temperature, and relative humidity). The concentrations of air pollutants were not normally distributed (data not shown), therefore we analyzed our data by nonparametric statistics, also known as distribution-free statistics. A paired Student's *t*-test was used to

assess the difference in the average concentration of air pollutants between indoor and outdoor, between window opening periods and reference periods, and between cooking periods and reference periods. With the objective of evaluating significant differences among the areas (general, traffic, and industry) for all air pollutants variables, data were analyzed using one-way analysis of variance (ANOVA) with Scheffe multiple comparison test. The generalized estimating equations (GEE) is a general statistical method in a longitudinal study with small samples for adjusting time interference, in which each time point is an independent event. Finally, the relationships between the 24-hour average concentrations of indoor air pollutants (dependent variable) and household characteristics (independent variable) were analyzed using GEE, adjusting for other household characteristics, and time interference. A *p*-value of less than 0.05 was considered significant.

## 3. Results

Table 1 shows the descriptive statistics of 24-h average indoor and outdoor air pollutants, temperature, and relative humidity in 60 houses. When indoor air pollutants were paired with outdoors within the same home, we found that the 24-hour average concentrations of indoor CO, $CO_2$, and $NO_2$ were significantly higher than the 24-hour average of outdoors concentrations, whereas, outdoor $O_3$ and $PM_{2.5}$ concentrations were significantly higher than indoor concentrations (all $p < 0.01$). The average distance between homes of subjects and their school were 0.86 km, 0.94 km, and 1.46 km in general, traffic, and industrial areas, respectively, as well as, the average distance between homes of subjects and the nearest air monitoring station were 1.07 km, 0.97 km, and 1.46 km in general, traffic, and industrial areas, respectively.

**Table 1.** Descriptive statistics of 24-h average indoor and outdoor air pollutants, temperature, and relative humidity in 60 houses.

| | | Mean | Median | Standard Deviation | Minimum | Maximum | *p*-Value [#] |
|---|---|---|---|---|---|---|---|
| CO (ppm) | indoor | 3.47 | 0.83 | 4.29 | 0.00 | 12.27 | 0.004 [‡] |
| | outdoor | 0.60 | 0.38 | 0.55 | 0.00 | 1.98 | |
| $CO_2$ (ppm) | indoor | 655.43 | 479.55 | 321.60 | 413.82 | 1320.00 | <0.001 [‡] |
| | outdoor | 322.22 | 319.92 | 17.23 | 285.83 | 353.90 | |
| $NO_2$ (ppb) | indoor | 185.30 | 177.97 | 41.52 | 127.28 | 251.41 | 0.008 [‡] |
| | outdoor | 107.54 | 118.22 | 36.83 | 39.90 | 149.80 | |
| $SO_2$ (ppm) | indoor | 0.00 | 0.00 | 0.00 | 0.00 | 0.00 | 0.193 |
| | outdoor | 0.01 | 0.00 | 0.02 | 0.00 | 0.06 | |
| $O_3$ (ppb) | indoor | 11.04 | 8.50 | 8.93 | 1.06 | 32.29 | 0.006 [‡] |
| | outdoor | 13.46 | 9.20 | 12.34 | 0.24 | 45.50 | |
| $PM_{2.5}$ ($\mu g/m^3$) | indoor | 60.00 | 40.00 | 50.00 | 10.00 | 210.00 | 0.001 [‡] |
| | outdoor | 110.00 | 90.00 | 90.00 | 30.00 | 410.00 | |
| Temperature (°C) | indoor | 31.00 | 31.00 | 1.76 | 26.00 | 34.00 | - |
| Relative humidity (%) | indoor | 72.00 | 72.00 | 4.98 | 62.00 | 84.00 | - |

[#] Paired Student's *t*-test, [‡] $p < 0.01$.

In comparison with household air pollutants of three areas, Table S1 shows descriptive statistics of 24-h average concentration of indoor air pollutants in the houses of traffic, industry, and general areas. Figure 1 shows the 24-h average concentration of indoor air pollutants (A) CO, (B) $CO_2$, (C) $NO_2$, and (D) $O_3$ in the houses of traffic, industry, and general areas. We found the 24-hour average concentration of indoor CO concentration of the traffic area was significantly higher than that of the industrial area, and the general area with all $p < 0.01$ (Figure 1, Table S1). In addition, the 24-hour average concentration of indoor $CO_2$ level of the general area was significantly lower than that of the traffic area, and industrial area (all $p < 0.01$) (Figure 1, Table S1). Finally, both the 24-hour average concentration of household $NO_2$ and $O_3$ concentrations of the industrial area were significantly lower than that of the traffic area, and general area (all $p < 0.01$) (Figure 1, Table S1). Moreover, there was no

statistical significant difference of the 24-hour average concentration of indoor $SO_2$ and $PM_{2.5}$ between the three areas (Table S1).

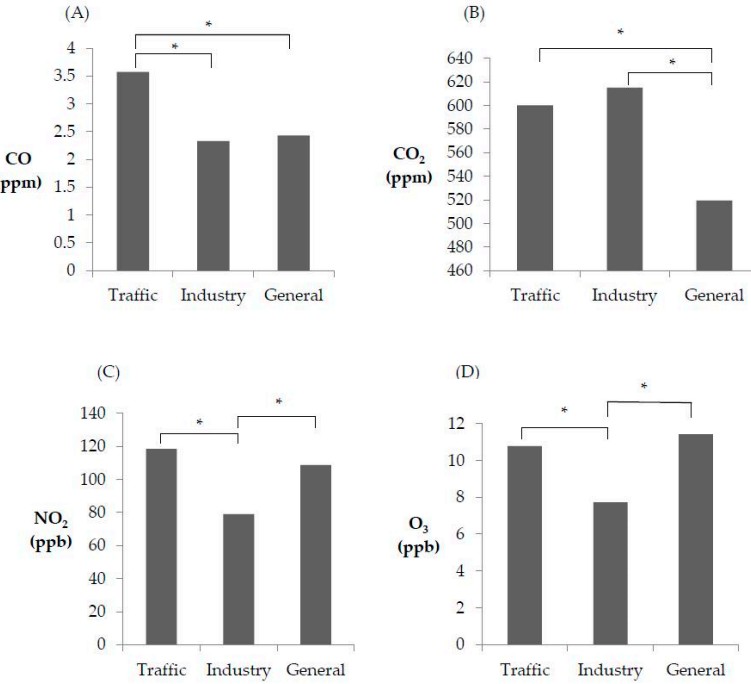

**Figure 1.** The 24-h average concentration of indoor air pollutants (**A**) CO, (**B**) $CO_2$, (**C**) $NO_2$, and (**D**) $O_3$ in the houses of traffic, industry, and general areas. * Scheffe test $p < 0.01$.

Table 2 shows the percentage of household characteristics including window opening, residents >4 people, cooking, etc. in traffic, industry, and general areas. We found compared with traffic area and industrial area, the general area had a higher percentage of window opening, cooking, and air-conditioning use; moreover, a lower percentage of residents > 4 people, smoker, incense burning, mosquito coil burning, and essential oil using.

**Table 2.** The percentage (%) of household characteristics in traffic, industry, and general areas.

|  | Area | | |
| --- | --- | --- | --- |
|  | **Traffic** | **Industry** | **General** |
| Window opening | 68.75 | 66.67 | 87.50 |
| Occupants (>4 people) | 40.40 | 57.01 | 34.41 |
| Cooking | 87.50 | 83.33 | 95.83 |
| Air-conditioning use | 62.50 | 83.33 | 95.83 |
| Making tea | 31.25 | 30.00 | 0 |
| Smoker | 63.64 | 40.00 | 26.09 |
| Incense burning | 72.73 | 50.00 | 29.17 |
| Mosquito coil burning | 37.50 | 22.22 | 12.50 |
| Essential oil using | 31.25 | 33.33 | 25.00 |

The following Table 3 shows the ratios of air pollutants during window opening periods to the reference periods and the differences in air pollutants between window opening periods and reference periods. The median ratios of pollutants during window opening periods to the reference periods for $NO_2$ and $PM_{2.5}$ were 1.56 and 1.13, respectively with the maximum values up to 5.23 and 1.85 respectively (Table 3). The $NO_2$ and $PM_{2.5}$ levels during window opening periods were significantly higher than that of the reference periods, and the maximum increased values were 53.25 ppb and 44 µg/m$^3$, respectively. Table 4 shows the ratios of air pollutants during cooking periods to reference

periods and the differences in air pollutants between window opening periods and reference periods. The median ratios of pollutants during cooking periods to the reference periods for CO, $CO_2$, $NO_2$, and $PM_{2.5}$ were 0.93, 1.06, 1.11, and 1.09, respectively. The concentrations of $CO_2$, $NO_2$, and $PM_{2.5}$ during the cooking periods were significantly higher than those of reference periods with increased concentrations of 26.17 ppm, 5.40 ppb, and 5 $\mu g/m^3$, respectively. However, the CO level during cooking periods was significantly lower than that of the reference periods with the decreased concentration of 0.25 ppm.

**Table 3.** The ratios of air pollutants during window opening periods to reference periods and the differences in air pollutants between window opening periods and reference periods.

| | Ratios (Window Opening Periods/Reference Periods [§]) | | | | Differences (Window Opening Periods − Reference Periods [§]) | | | | *p*-Value [#] |
|---|---|---|---|---|---|---|---|---|---|
| | Median | S.D. | Min. | Max. | Median | S.D. | Min. | Max. | |
| CO (ppm) | 0.98 | 1.34 | 0.57 | 4.44 | 0.00 | 1.31 | −2.42 | 3.67 | 0.53 |
| $CO_2$ (ppm) | 1.05 | 0.18 | 0.73 | 1.43 | 29 | 128 | −141 | 296 | 0.21 |
| $NO_2$ (ppb) | 1.56 | 1.30 | 0.94 | 5.23 | 18.71 | 16.05 | −9.40 | 53.25 | <0.01 [‡] |
| $SO_2$ (ppm) | 0.00 | 0.92 | 0.00 | 3.27 | 0.00 | 0.02 | 0.00 | 0.08 | 0.21 |
| $O_3$ (ppb) | 1.18 | 0.59 | 0.56 | 2.19 | 0.81 | 4.44 | −11.91 | 10.05 | 0.52 |
| $PM_{2.5}$ ($\mu g/m^3$) | 1.13 | 0.31 | 0.69 | 1.85 | 7 | 16.20 | −6 | 44 | 0.04 [†] |

[#] Paired Student's *t*-test, [†] $p < 0.05$, [‡] $p < 0.01$. [§] Reference periods were two one-hour periods before and after window opening periods.

**Table 4.** The ratios of air pollutants during cooking periods to reference periods and the differences in air pollutants between during cooking periods and reference periods.

| | Ratios (Cooking Periods/Reference Periods [§]) | | | | Differences (Cooking Periods − Reference Periods [§]) | | | | *p*-Value [#] |
|---|---|---|---|---|---|---|---|---|---|
| | Median | S.D. | Min. | Max. | Median | S.D. | Min. | Max. | |
| CO (ppm) | 0.93 | 0.22 | 0.46 | 1.51 | −0.25 | 0.84 | −3.53 | 0.61 | <0.01 [‡] |
| $CO_2$ (ppm) | 1.06 | 0.14 | 0.85 | 1.58 | 26.17 | 90.21 | −111.67 | 342.5 | <0.01 [‡] |
| $NO_2$ (ppb) | 1.11 | 0.98 | 0.51 | 5.43 | 5.40 | 29.71 | −71.17 | 101.75 | <0.01 [‡] |
| $O_3$ (ppb) | 1.08 | 0.69 | 0.46 | 4.36 | 0.27 | 8.89 | −35.14 | 17.08 | 0.94 |
| $PM_{2.5}$ ($\mu g/m^3$) | 1.09 | 0.30 | 0.60 | 2.56 | 5 | 14 | −45 | 56 | 0.04 [†] |

[#] Paired Student's *t*-test, [†] $p < 0.05$, [‡] $p < 0.01$. [§] Reference periods were the one-hour period before cooking periods.

Table 5 shows the association between air pollutants concentrations (24-h average concentration of air pollutants in each house as dependent variable), and household characteristics by using the generalized estimating equations model. This study revealed that CO concentrations were positively associated with the number of occupants, cleaning, smoking, incense burning, mosquito coil burning, and negatively correlated to cooking with a statistical significance. Indoor $CO_2$ concentrations were positively associated with the number of occupants, air-conditioning use, smoking, incense burning, and negatively correlated to mosquito coil burning with a statistical significance. In addition, significantly higher $NO_2$ levels were found in the homes with smokers than homes without smokers. There were significantly positive associations between indoor $SO_2$ concentrations and smoking and incense burning. In terms of $O_3$, indoor $O_3$ concentrations were positively associated with the window opening and negatively correlated to the number of occupants, incense burning, and essential oil use with a statistical significance. For $PM_{2.5}$, it was positively associated with cleaning and incense burning with a statistical significance.

**Table 5.** Association between air pollutants concentrations (24-h average concentration of air pollutants in each house as dependent variable), and household characteristics: generalized estimating equations.

| | CO (ppm) | $CO_2$ (ppm) | $NO_2$ (ppb) | $SO_2$ (ppm) | $O_3$ (ppb) | $PM_{2.5}$ (µg/m³) |
|---|---|---|---|---|---|---|
| Window opening (Yes vs. No) | 0.32 | 84.84 | −0.61 | 0.44 | 24.34 [‡] | −0.021 |
| Occupants | 0.52 [‡] | 51.62 [‡] | 3.02 | −0.008 | −3.49 [‡] | 0.004 |
| Cleaning (Yes vs. No) | 4.73 [†] | −317.49 | 1.39 | 0.43 | −6.24 | 0.047 [†] |
| Cooking (Yes vs. No) | −3.89 [†] | 228.02 | −28.01 | −0.21 | 1.79 | 0.065 |
| Fan using (Yes vs. No) | 1.42 | −32.97 | 10.58 | −0.0003 | −2.07 | 0.002 |
| Air- conditioning use (Yes vs. No) | −1.22 | 246.99 [‡] | 87.87 | 0.25 | 21.59 | 0.008 |
| Making tea (Yes vs. No) | 37.04 | - | −0.45 | −0.13 | 14.21 | −0.050 |
| Smoking (Yes vs. No) | 17.21 [†] | 1988.44 [‡] | 547.36 [‡] | 2.98 [‡] | 1.69 | 0.173 |
| Incense burning (Yes vs. No) | 18.21 [†] | 2927.87 [†] | 193.11 | 3.66 [‡] | −108.9 [‡] | 0.416 [‡] |
| Mosquito coil burning (Yes vs. No) | 41.55 [‡] | −892.64 [†] | 673.52 | 2.67 | 2.29 | - |
| Essential oil use (Yes vs. No) | 12.76 | 269.25 | 74.89 | −0.66 | −89.29 [‡] | −0.022 |

Generalized estimating equations (GEE) [†] $p < 0.05$, [‡] $p < 0.01$.

## 4. Discussion

Our results showed that the outdoor concentrations of $O_3$ and $PM_{2.5}$ were significantly higher than indoor concentrations. The Kaohsiung City is a city with intense traffic and heavy industries, and previous studies believed $SO_2$, $NO_x$, $PM_{2.5}$, and CO were the major conventional air pollutant in steel plants, oil refineries, and vehicular exhaust emissions [12–14,24]. In addition, outdoor $O_3$ might be formed by the photochemical reaction of nitrogen oxides absorbing sunlight, and VOCs [25,26]. According to the PSI database from 2010 to 2012 of Taiwan EPA, only $O_3$ and total suspended particulate (TSP) would exceed the standard [27]. This may be the reason why outdoor $PM_{2.5}$ and $O_3$ concentrations were higher than indoor concentrations. In our study, outdoor median $PM_{2.5}$ levels (90 µg/m³) were higher than both the National Ambient Air Quality Standards of UAS and Taiwan EPA with the 24-hour standard for $PM_{2.5}$ of 35 µg/m³. In addition, the median value of indoor $PM_{2.5}$ concentrations (40 µg/m³) was also higher than the criteria of indoor air quality (IAQ) standards of Taiwan EPA (35 µg/m³/24 h). In our study, indoor CO, $CO_2$, and $NO_2$ levels were significantly higher than outdoor levels. The number of occupants and human activities such as cooking, smoking, etc. might be the factors affecting indoor pollutants whereas liquefied petroleum gas (LPG), not electric stoves, was the main cooking way in Kaohsiung City [28,29]. In addition, most of the houses were just by the roads and very close to the mobile sources in Kaohsiung City, which was thought of as a traffic-intensive city with the number of cars and motorcycles of approximately 430,000 and 1,230,000, respectively, in 2010 [30]. Thus, the main combustion products of vehicular engines such as CO, $NO_X$, etc. entering the houses through cracks and windows might be the reason why the indoor concentrations of CO, $CO_2$, and $NO_2$ were higher than outdoor concentrations [15,16].

In comparison with traffic, industrial, and general areas, the highest household CO concentration was found in the traffic area among the three areas. According to the previous study, the greatest source of CO (more than 90%) in cities was motor vehicles [24]. The high traffic flow in the traffic area might be the reason for the observation. For $CO_2$, our study indicated that the lowest household $CO_2$ level was in the general area among the three areas. The main source of $CO_2$ was from human respiration [24,31]. The number of residents might be one possible reason since the number of residents > 4 people in the traffic area, the industrial area, and general area were 40.40%, 57.01%, and 34.41%, respectively. We also found both household $NO_2$ and $O_3$ concentrations of the industrial area were

lowest among the three areas, which was not consistent with the observations of previous studies that ambient $NO_2$ was related to industrial activities [24], and outdoor $O_3$ might be formed by the photochemical reaction of nitrogen oxides absorbing sunlight, and VOCs [25,26]. We believed these may be related to Taiwan EPA's policies and efforts to control air pollution from stationary sources after that the "Stationary Pollution Source Air Pollutant Emissions Standards" was passed in 1992, and the "Air Pollution Control Act Enforcement Rules" was also implemented in 2003.

In regard to the effects on the window opening, our study displayed that household $NO_2$ and $PM_{2.5}$ concentrations during window opening periods were significantly higher than that during reference periods. $NO_X$ and PM were related to traffic emissions [24,32], and most of the houses in Taiwan were adjacent to roads, so window opening might increase indoor $NO_2$ and $PM_{2.5}$. For the influence of cooking, there were many simulated experiments exploring the air pollutant emissions of cooking-related fuel combustion [29,33–36], and they demonstrated that CO, $CO_2$, $NO_X$, and $PM_{2.5}$ would be emitted by the process of the experiments. Although CO also was produced by cooking, it was revealed that combustion of high-grade fuels (such as natural gas, and LPG which contained propane, butane, etc.), the main fuel-burning stoves use in Taiwan households usually produce much less CO than combustion of low-grade fuels [29,33]. In the previous study, Delp et al. revealed the residential cooking exhaust hoods could not completely capture the pollutants and their efficiency was highly variable [37]. Our results showed that indoor $CO_2$, $NO_2$, and $PM_{2.5}$ levels during cooking periods were significantly higher than during reference periods, but the indoor CO level during cooking periods was lower than during reference periods, possibly indicating that the emission rate of $CO_2$, $NO_2$, and $PM_{2.5}$ might be higher than the capture rate of the exhaust hood and the emission rate of CO might be lower than the pollutants capture rate of the exhaust hood.

In terms of influence factors, we found there were significantly positive correlations between the number of occupants and CO and $CO_2$ concentrations. Our study was consistent with the observations of the previous study that $CO_2$ was produced by human respiration [24,31]. In addition to the combustion, the indoor CO also was related to the status of residents; the previous studies revealed either a smoking person or person with inflammatory diseases exhaled higher CO levels than control group [38,39]. We also found smoking was significantly positively associated with household CO in our study. According to previous studies, smoking, incense burning, and mosquito coil burning were significantly positively associated with CO, $CO_2$, $SO_2$, $NO_X$, and PM [40–42], and these results were consistent with our observation. The cleaning behavior would increase indoor $PM_{2.5}$ and CO levels; it was consistent with the previous study that indoor $PM_{2.5}$ and $PM_5$ levels could be elevated by the cleaning behavior of dry dust, and vacuuming [43]. In addition, commercial cleansers and disinfectants contain VOCs [44], and El Fadel et al. found VOCs concentration was positively correlated with CO concentration [45]. We also revealed that air-conditioning use was positively associated with indoor $CO_2$ concentrations with a statistical significance, which was consistent with a previous observation that $CO_2$ levels were higher in mechanically ventilated buildings than in naturally ventilated buildings [46]. There was a significantly negative association between essential oil use and $O_3$ concentration. The commercially available essential oils contain many VOCs (e.g., D-limonene, $\alpha$- pinene, etc.) [47], in addition, a study displayed that indoor VOCs level had increased significantly after burning essential oils [48]. $O_3$ was one of the indoor oxidants [49,50], and Waring et al. demonstrated that 68% of all $O_3$ reactions were with D-limonene, and 26% of all $O_3$ reactions are with $\alpha$-pinene [50]. This might be the reason why the essential oil use could decrease the $O_3$ level. Finally, by questionnaire, it was found that window opening was significantly correlated with increased $O_3$ concentration, which was not consistent with the results from the time–microenvironment–activity-diary that only $NO_2$ and $PM_{2.5}$ levels during the window opening periods were significantly higher than that of reference periods. We believed $O_3$ was a major component of photochemical pollution, so it is more relevant to outdoor sources than indoor sources. Thus, compared with the households which closed the windows, the households which opened the windows had a significantly higher 24-hour average concentration of $O_3$. When comparing the window opening periods with the reference periods (two one-hour periods before and after window

opening periods), there was no significant variation in atmospheric $O_3$ concentration in a short time (within three hours). For $PM_{2.5}$ and $NO_X$ levels, there was no significant difference between households which closed and opened the windows, the possible reason might be that $PM_{2.5}$ and $NO_X$ could come from both indoor (cooking) and outdoor (traffic) sources.

## 5. Conclusions

This study explored the concentration of indoor air pollutants in different areas including traffic, industrial, and general areas within an industrial city. Moreover, this study also revealed household $NO_2$ and $PM_{2.5}$ concentrations during window opening periods were significantly higher than that of the reference periods with increased concentrations of 18.71 ppb, and 7 μg/m$^3$, respectively. For the influence of cooking, indoor $CO_2$, $NO_2$, and $PM_{2.5}$ levels during the cooking periods were significantly higher than that of the reference periods with increased concentrations of 26.17 ppm, 5.40 ppb, and 5 μg/m$^3$, respectively.

**Supplementary Materials:** The following are available online at http://www.mdpi.com/2076-3417/9/20/4306/s1, Table S1: Descriptive statistics of 24-h average concentration of indoor air pollutants in the houses of traffic, industry, and general areas.

**Author Contributions:** Conceptualization, C.-Y.Y.; formal analysis, Y.-C.Y.; writing—original draft preparation, Y.-T.C. and Y.-T.C.; writing—review and editing, P.-S.C. and K.D.M.

**Funding:** This research was funded by "Wang Jhan-Yang Public Trust Fund", grant number 108-002-6.

**Conflicts of Interest:** The authors declare no conflict of interest.

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
