# Peer review of "Cooking/Window Opening and Associated Increases of Indoor PM2.5 and NO2 Concentrations of Children’s Houses in Kaohsiung, Taiwan"

_applsci, doi:10.3390/app9204306_

Round 1

Reviewer 1 Report

The paper require the modifies reported in yellow in the attached file

Author Response

Thank you very much for your suggestion.

Reviewer 2 Report

After reading your article, I have some doubts concerning its publication mainly due to:

it is a study from 2010; the areas (traffic, industry and general) are not well-defined; it should be mentioned the difference between the smokers' houses for each area, for instance;  the same should have been done for each "air pollutants concentrations and household characteristics" for each area; "window opening period" corresponds to which time of the day: rush hour, during day, while cooking, ...? Was it the same for each house?; and the reference period?; although the time period (24h) is short, was it during weekdays for every house?

As you may observe, there are a lot of questions about your study.

Author Response

(The authors gave the same response as above.)

Reviewer 3 Report

although the research area of this work is quite interesting, the presentation of the methodological framework, results and discussion have serious criticisms. Another important issue is the English syntax and grammatical patterns, which make the reading of th manuscript difficult to follow. The innovation of the study should be clearly stated at the introduction. A lot of details and clarifications are missing: time frames,  characteristics of each area, time intervals of measurements and activities. Overall, I wouldn't recommend the publication of this manuscript at this form, but i would encourage the authors for revising and re-submitting it in future.

Author Response

(The authors gave the same response as above.)

Reviewer 4 Report

General comment

The manuscript presents the results the comparison of CO, CO2, NO2, SO2, O3, PM2.5, and their influence factors window opening, a number of occupants, cleaning, cooking, fan using, air-condition using, tea making, smoking, incense burning, mosquito coil burning, and essential oil burning.

The article could be interesting for the readers however it demands some improvement.

The introduction should include the general information about CO, CO2, NO2, SO2, O3, PM2.5  concentrations in ambient and indoor air in the region or country.

The concentrations of SO2 are 0.00 please explain the determination range, and detection limit.

Instead of some Tables (which can be put in supporting material), it would be better to present the results on the Figures.

The reference period should be described in detail.

Please describe the GEE model.

Conclusions too general.

References 10, 37, 38, 43, 45, 47 need the proper format.

Specific comments

The beginning of the introduction and abstract are the same.

Line 36 double space

Line 44 concentration should be deleted

Line49 utilized seams strange

Line 69 on the roof?

Line 70 monitoring station

In many lines, there is a space mg/ m3 or mg/ m3 which should be deleted

In a few lines is P-value instead of the p-value

The p-values between industry and general? Or traffic and general (lines 145-146)

Line 158 cooking period /reference (period?) ratios

Lines 158-163 language

Table 4 where is window opening and reference period.

In a few lines, there is a full name of volatile organic compounds it should be VOCs

Lines 236 and237 during is missing

Line 262, 271,272 language

Author Response

(The authors gave the same response as above.)

Round 2

Reviewer 2 Report

After reading the corrections and the improved version, the article can be send for publication.

Author Response

Thank you very much.

Reviewer 3 Report

Dear authors, 

thank you for your responses. The manuscript has been improved in an adequate level. However, I insist in a further and stricter grammatical and syntax error checking, as there is a number of problematic expressions e.g. air qualitiES (LINE 28); peopleS (LINE 177). 

Author Response

air qualitiES (LINE 28) and peopleS (LINE 177) were modified.

A further grammatical and syntax error checking was done for the whole manuscript.

Thank you very much for the suggestion.

Reviewer 4 Report

I appreciate that the authors followed my comments, and I confirm that the article is revised according to my suggestions.

Author Response

Thank you very much.